# Impact of the Structural Modifications of Potato Protein in the Digestibility Process under Semi-Dynamic Simulated Human Gastrointestinal In Vitro System

**DOI:** 10.3390/nu14122505

**Published:** 2022-06-16

**Authors:** Luis Jiménez-Munoz, Emmanouil D. Tsochatzis, Milena Corredig

**Affiliations:** 1Department of Food Science, CiFOOD Center for Innovative Foods, Aarhus University, Agro Food Park 48, 8200 Aarhus, Denmark; emmanouil.tsochatzis@efsa.europa.eu (E.D.T.); mc@food.au.dk (M.C.); 2European Food Safety Authority-EFSA, Via Carlo Magno 1A, 43146 Parma, Italy

**Keywords:** potato protein isolate, patatin, semi-dynamic in vitro digestion, structuring, amino acid digestibility

## Abstract

The raising consumer demand for plant-derived proteins has led to an increased production of alternative protein ingredients with varying processing histories. In this study, we used a commercially available potato protein ingredient with a nutritionally valuable amino acid profile and high technological functionality to evaluate if the digestibility of a suspension with the same composition is affected by differences in the structure. Four isocaloric (4% protein, *w*/*w*) matrices (suspension, gel, foam and heat-set foam) were prepared and their gastrointestinal fate was followed utilizing a semi-dynamic in vitro digestion model. The microstructure was observed by confocal laser scanning microscopy, protein breakdown was tested by electrophoresis and free amino acids after intestinal digestion was estimated using liquid chromatography/triple-quadruple-mass spectrometry (LC-TQMS). The heat-treated samples showed a higher degree of hydrolysis and lower trypsin inhibitory activity than the non-heat-treated samples. An in vitro digestible indispensable amino acid score was calculated based on experimental data, showing a value of 0.9 based on sulfur amino acids/valine as the limiting amino acids. The heated samples also showed a slower gastric emptying rate. The study highlights the effect of the food matrix on the distribution of the peptides created during various stages of gastric emptying.

## 1. Introduction

Plant proteins are currently of great interest as they have shown great potential to be used as ingredients in more sustainable and climate friendly foods compared to their traditional animal-derived counterparts [1]. However, more research is needed to understand how to design and offer novel food matrices containing plant proteins able to respond to consumers’ demands for wholesome and nutritionally functional foods. Necessarily, a great part of global research so far has focused on the study of extraction and concentration processes aiming to increase the purity of ingredients, to decrease antinutritional components, off-flavor precursors, to improve their technological performance and their nutritional profile. Isolates (≥80% protein) can be obtained with various technologies [2] and each process will result in different effects on the purity, functionality, presence and activity of anti-nutritional factors. Furthermore, protein ingredients, once processed in a food, may have different properties which will affect their digestibility, peptide distributions during gastrointestinal transit, and nutritional bioefficacy [3,4,5].

Among surging plant-based alternatives, potato protein isolates stand out, as they have the potential to be used as a source of high quality protein for human consumption. Furthermore, their recovery from one of the side-streams of the production of potato starch creates an opportunity to valorize and improve the circularity of this food chain [6]. There are four major classes which compose potato protein extract. Patatin (40 kDa), a glycoprotein (up to 40%), and protease inhibitors (22, 15, 9 kDa) represent up to ~90% [7] of the total extract, with lipoxygenase (130 kDa) making up the rest. The complex trypsin inhibitor group consist mainly of PI I/II, PKPI, PCI, A5-inhibitor and Bowman Birk inhibitor (BBI) [8]. Prior research has suggested that potato peptides have biological activities. For example, they have been shown to regulate blood pressure and serum cholesterol [9] and have ACE-inhibitory and antioxidant properties [10]. Potato trypsin inhibitor (PTI) fractions may also be of great interest, as they are reported to increase satiety levels, reduce food intake and delay gastric emptying by elevating Cholecystokinin levels in plasma [11,12].

Conventionally, the industrial production of the protein fraction from potato juice is achieved by applying isoelectric precipitation or heat coagulation, after the extraction of the starch. In these processes, despite the high yield achieved, modifications in the protein structure due to harsh conditions lead to decreases in functionality compared to the native isolate, resulting in ingredients mainly suitable for animal feed [13]. To overcome this challenge, technologies such as ammonium sulfate ((NH_4_)_2_SO_4_) precipitation, membrane separation (micro, ultra and nano-filtration), ion exchange (IEX) and expanded bed adsorption (EBA) chromatography have been applied as more gentle approaches in the recovery and isolation of potato protein [14]. Furthermore, using these processes, based on the intrinsic physical differences between the proteins in the extract, it is possible to remove antinutritional factors such as glycoalkaloids (chaconine and solanine). These processes can result in an ingredient of high technological functionality [15]. The isolates obtained still maintain high solubility and allow the design of various structures in foods, due to their gelling, emulsifying and foaming properties.

In this study, it was hypothesized that potato protein digestibility is affected by its processing history when used as an ingredient, ultimately affecting its biological functions. For this reason, different structures were prepared using potato protein isolates and were subjected to different in vitro digestion kinetics. Although animal and human intervention trials are still the gold standard to measure protein digestibility, these tests are ethically challenging, and for this reason, in vitro methods are often preferred to provide a mechanistic understanding of the effect of the food matrix on digestion behavior. To test this hypothesis, a commercial potato protein ingredient (AVEBE, Solanic^®^ 200) [16], with high functionality (solubility, foaming and gelling capacity) was used to prepare different matrices (suspension, gel, unstable and stable foam) of isocaloric content. These systems were then subjected to a semi-dynamic in vitro digestion study, a method that takes into consideration the structural changes occurring to food in the stomach, and provides the necessary conditions to understand the fate of the food matrix in the gastrointestinal tract. This was in order to determine how the structures formed before and during gastrointestinal transit may play a role in their digestibility.

## 2. Materials and Methods

### 2.1. Preparation of Potato Protein Structures

A commercially available, patatin-rich potato protein isolate (PoPI) (Solanic^®^ 200, Avebe, Veendam, The Netherlands) was used in this study. The manufacturer’s specification indicated that PoPI contained 87.5% protein (as is), 3.5% fiber (as is), 0.2% carbohydrates (as is), 0.2% fat (as is), 5% moisture content and 4% ash (as is). Four different structures containing the same amount of protein (4% protein, *w*/*w*) were prepared: a suspension (PoPI-S), a heat induced gel (PoPI-G), a foam (PoPI-F) and a heated foam (PoPI-HF). PoPI-S was prepared by dissolving PoPI powder in deionized water and stirring overnight at room temperature. As per manufacturer’s specification, the protein showed ≥85% solubility. PoPI-G was instead prepared by heating the suspension at 90 °C for 15 min in a water bath. This caused the formation of a gel. PoPI-F was a foam prepared by whipping the suspension for 3 min in a Thermomix^®^ (Vorwerk) (Wuppertal, Germany) (using the butterfly whisk at speed #3). The fourth sample (PoPI-HF) consisted of a heated foam structure produced by treating the previously mentioned foam in a microwave (Siemens^®^, BF525LMS0, Aarhus, Denmark) at 360 W for 1 min. The temperature inside the heat-set foam was 72 °C, as measured immediately after heating using a thermocouple (Testo 100^®^, Titisee-Neustadt, Germany).

### 2.2. Determination of Trypsin Inhibitory Activity in PPI

The residual trypsin inhibitor (TI) activity was determined as previously published [17]. Trypsin from porcine pancreas (T4799) was dissolved in 1 mM HCl solution (pH 3.0) to a concentration of 500 μg/mL. Nα-Benzoyl-l-arginine 4-nitroanilide hydrochloride (BApNA) dissolved in Tris-buffer solution (400 μg/mL) (0.05 M, pH 8.1) was used as substrate, and 30% acetic acid (*v*/*v*) was used to stop trypsin activity. The blank was prepared by adding 1 mL of acetic acid and 2 mL of trypsin solution; to this mix, 2 mL of deionized water and 5 mL of BApNA solution were subsequently added. For the samples/control: 2 mL of trypsin solution was mixed with 2 mL of PoPI samples, or with deionized water for the control. Once mixed, the tubes were submerged in a water bath (37 °C, 10 min). Subsequently, 5 mL of BApNA substrate solution was added, and incubated for 10 min. Finally, 1 mL of 30% acetic acid was added to stop the reaction. The samples and controls and blank were transferred to 1 × 1 cm cuvettes and the absorbance of the resulting product measured 410 nm using a Cary 60 Bio UV-spectrophotometer. A decrease in 0.01 U of absorbance in the assay (15 min, 37 °C) was defined as one unit of trypsin inhibitory activity (TIA). The extract was diluted to reach TI (%) values ranging between 25–70%. The residual trypsin inhibitory activity and the residue percentage were calculated using the following equations:$$\mathrm{TIA}\;(U)=\frac{\left(A_{c}-A_{0})-(A_{t}-A_{0}\right)\times N}{0.01}$$
$$\mathrm{TIA}\;(\%)=\frac{\left(A_{c}-A_{0}\right)-\left(A_{t}-A_{0}\right)}{\left(A_{c}-A_{0}\right)}\times 100\%$$
where *A_c_* is the absorbance of the control, *A_t_* is the absorbance of the sample, *A*_0_ is the absorbance of the blank, and N is the dilution factor.

### 2.3. Gastrointestinal Simulated Semi-Dynamic in Vitro Digestion

A standardized semi-dynamic in vitro digestion model [18] was used with the purpose of simulating the environment during gastrointestinal transit. The protein hydrolysis was carried out under physiologically relevant conditions (increasing enzyme concentrations, changes in the pH and emptying dynamics). Electrolyte simulated salivary fluid (eSSF), gastric fluid (eSGF) and simulated intestinal fluid (eSIF) were prepared according to INFOGEST 2.0 protocol, as well as their mixing ratio with food and enzymes [19]. All enzymatic activities were also measured according to the protocol. Reagents were of analytical grade.

The caloric content of the four different PoPI model foods was calculated according to the Atwater factor (4 kCal g^−1^ for protein, 4 kCal g^−1^ for carbohydrates and 9 kCal g^−1^ for lipids) based on proximate composition of the sample. The gastric emptying time was estimated based on data of in vivo emptying dynamics (2 kCal min^−1^ emptying of 500 mL food), and this time was also used to determine the rate of addition of the simulated fluids and enzymes [20]. Given the low content of carbohydrates and lipids in PoPI powder, salivary amylase and gastric lipase were omitted in the in vitro experiment. Based on the sample size (10 mL, 1.65 kcal) of the current work, the duration of gastric digestion was 41.3 min. Gastric aliquots (3.82 mL) were collected every 8.26 min, for five gastric emptying points numbered in sequence (GE1–GE5). The total amount of 0.5 M HCl needed for gastric digestion was estimated by testing the volume needed to reach a final pH ~2. Porcine pepsin (2000 U mL^−1^) (Sigma Aldrich, 3760 V, 3300 U mg^−1^, Darmstadt, Germany) was added at a secretion rate of 48.42 μL min^−1^. Immediately after collection, the pH of the gastric samples was raised to 7 by adding NaOH (0.5 M), so pepsin activity was inhibited. Subsequently, the emptied gastric samples were subjected to static intestinal digestion [19,21]. The digesta, from the five emptying points (GE1–GE5), were mixed in falcon tubes for two hours at 37 °C in an incubator with eSIF with bovine bile extract (10 mM bile salts) (Sigma Aldrich, B8631, Darmstadt, Germany) and pancreatin (200 U mL^−1^, trypsin activity in eSIF) (from porcine pancreas, Sigma Aldrich, P1750-25G, Darmstadt, Germany). To stop trypsin and chymotrypsin activity, 1.91 mL of 0.01 M Pefabloc^®^ (Darmstadt, Germany) was added. The intestinal endpoints, indicated as IE1–IE5, correspond to the previous gastric emptying points (GE1–GE5). Samples were immediately frozen at −80 °C for 24 h and then stored at −20 °C.

### 2.4. Confocal Laser Scanning Microscopy (CLSM)

Confocal laser scanning microscopy (CLSM) (Nikon Eclipse Ti series, Nikon Instruments Inc., Amsterdam, The Netherlands) was used to observe the microstructure of the four potato protein samples emptied at the different stages of gastrointestinal digestion. Aliquots of 1.5 mL were stained with 30 μL of 1 mg/mL fluorescein-5-isothiocyanate (FITC). The pinhole diameter was maintained at 1 Airy Unit. Images were taken at 20× dry objective and 60× water immersion objective, with a refractive index of 1.0 and 1.33, respectively. A 488 nm helium/neon laser was used and the filter was set at 475–650 nm to image labelled proteins.

### 2.5. Sodium Dodecyl Sulphate-Polyacrylamide Gel Electrophoresis (SDS-PAGE)

Protein breakdown during gastric and intestinal digestion was followed using SDS-PAGE. Samples (13 μL) of undigested material, the 5 gastric and intestinal endpoints, were mixed with 5 μL of NuPAGE^®^ LDS sample buffer and 2 μL NuPAGE reducing agent (Thermofisher, Waltham, MA, USA), and heated at 90 °C for 5 min. Samples were loaded on NuPAGE^®^ SDS-PAGE bis-tris gels (4–12%). Running buffer was prepared by mixing 50 mL NuPAGE^®^ MES-SDS Running Buffer (20×) with 950 mL deionized water. The separation process was run for 45 min at 200 V. Protein bands were stained with SimplyBlue (Invitrogen, Thermo Fisher Scientific, San Jose, CA, USA) and images were analyzed using ChemiDoc XRS+ and Image Lab software (Bio-Rad Lab., Hercules, CA, USA).

### 2.6. Size Exclusion Chromatography

Size exclusion chromatography (SEC-HPLC) was performed for the gastrointestinal points (GE1-GE5, I1 and I5 at 120 min) of the four different digestion treatments. The impact of the different treatments on the pattern formation of peptide aggregates in the soluble phase in different stages of gastrointestinal digestion was determined by analyzing the size distribution profile from the digesta. The analysis was carried out with a HPLC system (Agilent Technologies 1100 series) equipped with a TSKgel column G2000SWXL (7.8 mm × 600 mm) (Tosoh Corp., Merck, Germany). The digesta were centrifuged at 10,000× *g* for 30 min at room temperature, and the supernatant, corresponding to the bioaccessible fraction [22], was diluted 1:4 (sample: acetonitrile 30%, *v*/*v*) and filtered using a syringe filter (0.45 μm PVDF membrane, Gilson Scientific Ltd., Luton, UK). The separation was obtained isocratically with a mobile phase consisting of 0.1% (*v*/*v*) trifluoroacetic acid and 30% (*v*/*v*) acetonitrile, at a flow rate of 0.5 mL min^−1^, measuring the absorbance at 214 nm. Integration of the chromatograms was obtained using Agilent Chemstation software (Agilent Technologies, Santa Clara, CA, USA). To estimate the molecular weight distribution of the peptides present in the bioaccessible fraction, their elution time was compared to the elution time of standards: bovine serum albumin (67 kDa), carbonic anhydrase (29 kDa), β-lactoglobulin (18.4 kDa), aprotinin (6.5 kDa), and the amino acids histidine-leucine (0.268 kDa), phenylalanine (0.165 kDa) and glycine (0.075 kDa).

### 2.7. Degree of Hydrolysis

The degree of hydrolysis was measured using the o-phthaldiadehyde (OPA) assay. The digested samples (0.5 mL) were mixed with Trichloroacetic acid (0.83 mL) (TCA) (3.12% *w*/*w*, final concentration) to precipitate insoluble protein. Samples were then centrifuged at 10,000× *g* for 30 min at room temperature. Thereafter, the supernatant was filtered with a 0.45 μm filter (PVDF membrane, Gilson Scientific Ltd.) (Luton, United Kingdom) and mixed with a reagent mixture containing ultrapure water (88.1%, *w*/*w*), sodium tetraborate (3.81%, *w*/*w*), SDS (0.1%, *w*/*w*), Dithioerythritol (DTE) (0.088%, *w*/*w*), and OPA (0.08%, *w*/*w*), previously dissolved in 4 mL ethanol. Samples/standards (10 μL) were mixed with 250 μL of OPA reagent and the reaction was allowed to proceed for 15 min. Free NH_2_ groups values were determined subsequently using a spectrophotometric assay and the absorbance was recorded at 340 nm using a plate reader. A stock solution of 2 mg/mL of L-leucine (131.17 g/mol) was used to prepare the calibration curve with ranging concentrations from 0 to 15 μM. Each measurement was performed in triplicate. The degree of hydrolysis was calculated as a ratio of the number of hydrolyzed peptide bonds divided by the number of total peptide bonds present [23].

### 2.8. Free Amino Acid Analysis by Triple Quadrupole Mass Spectrometry

To estimate the protein digestibility, the amount of free amino acids in the supernatant of the intestinal endpoints (I5) was calculated using a triple quadrupole mass spectrometer (TQMS). To obtain the soluble fraction, the stated procedure in SEC-HPLC section was followed. Once centrifuged and filtered, 20 μL of sample was diluted with 180 μL of 0.1 M HCl, and 50 μL of internal standard consisting of stable isotopically labelled amino acids [“Cell Free” Amino acid mix (20 AA, U-13C, 97–99%; U-15N, 97–99%) (Cambridge Isotope Laboratories, Cambridge, MA, USA)] was added to the mixture. To prepare the stock solution of the internal standard, 0.1 M HCl was used and kept in separate vials at −20 °C. The samples were then subjected to a chromatographic separation on a UHPLC system (Agilent Technologies, 1290 Infinity II), using an Intrada Amino Acid column, 150 × 3 mm, 3 μm (Imtakt, Portland, ORE, USA) in combination with a Van-Guard Pre-column ACQUITY UPLC BEH HILIC, 5.0 × 2.1 mm, 1.7 μm (Waters, Ireland), at a flow rate of 0.6 mL/min. The analytical column was thermostatted at 35 °C (±0.8 °C) and the injection volume was 5 μL. Detection and mass analysis were performed on a Triple Quadruple Mass spectrometer (TQMS) system (6495 Agilent Technologies, Santa Clara, CA, USA) using MassHunter^®^ software (Agilent Technologies, Santa Clara, CA, USA) to interpret the data. Acetonitrile with formic acid (0.1%, *v*/*v*) and ammonium formate were used as solvents for the separation in the chromatographic column. A gradient elution program was used going from 20% (ammonium formate, 100 mM) (0 min) to 100% (12 min), followed by an isocratic elution of 100% ammonium formate for 2 min. The column was then re-equilibrated for 15 min with 20% ammonium formate after each run.

The reagents used in the mobile phase were the following: Acetonitrile, LC-MS grade (Merck, Steinheim, Germany); Ultrapure water (18.2 MΩ) (MilliQ, Millipore, Darmstadt, Germany) used in the preparation of solutions; formic acid, LC-MS grade (Sigma Aldrich, Steinheim, Germany) and Ammonium formate (Sigma Aldrich, Steinheim, Germany).

### 2.9. Statistical Analysis

The measurements reported for protein particle size, degree of hydrolysis, and trypsin inhibitory activity are the results of three independent measurements, while two independent measurements were used to estimate the molecular weight distribution and free amino acids. Statistical analysis was performed using IMB SPSS v22 (Armonk, IBM Corp, New York, NY, USA), and minimum significance was set at the 5% level (*p* < 0.05). One way analysis of variance (ANOVA) with Tukey HSD was applied to trypsin inhibitory activity and free amino acids and was conducted to determine the significance of the treatments, while a two-way multivariate analysis of variance (MANOVA) was applied to SEC-HPLC Mw distribution.

## 3. Results and Discussion

### 3.1. Microstructural Characterization

The visual appearance of the four structures prepared from the potato protein isolate is shown in Figure 1. Representative microstructure images corresponding to each of the four different potato protein treatments (PoPI-S, PoPI-G, PoPI-F, PoPI-HF) are presented in Figure 2. Images depict, for the same treatment, the initial samples (U), sequential stages of gastric emptying (G1, G3 and G5) and the final intestinal stage (I5). The latter represents the most extensively digested sample after simulated intestinal digestion. Figure 2 shows visible differences between the colloidal structures of the four matrices, before and throughout the in vitro digestive process. Images at higher magnification can be also found in the Appendix A.

The untreated dispersion (PoPI-S) was mostly homogeneous with a few small particles in suspension (Figure 2A). These particles represent residual insoluble powder sized between 10 and 50 μm. The apparent average diameter of these suspensions was about 20 μm when measured by integrated light scattering (data not shown). These results are well aligned with the known solubility of this commercial isolate. During gastric transit, the suspension showed the formation of aggregates. At the acidic pH of the gastric stages G1 and G3 (pH 5.5–3.6, Figure 2b,c), the protein suspensions were characterized by the presence of interacting particles, and at G3 there seemed to be a semi-continuous matrix, which could be attributed to the formation of an acid-induced network structure. It is indeed reported that protein-protein interactions occur as the surface charge of proteins get closer to their isoelectric point (4.5–5.1), leading to aggregation of the globular patatin [24]. At the final stage of in vitro gastric digestion (Figure 2d) at pH 2.0 (G5), the network is no longer visible. The residual aggregates, less than 50 μm in size, are diluted in the gastric juices, and are characterized by an increased charge repulsion, as the environment is now far from the isoelectric point of the protein particles [25]. The persistence of globular particles at this point would suggest an incomplete hydrolysis by pepsin at these gastric emptying times. Figure 2e depicts the microstructure of the final intestinal endpoint (I5) of PoPI-S. At this stage, the suspension shows extensive digestion, with only a few aggregates present, also deriving from the inhomogeneous suspension of the simulated in vitro intestinal juice.

Figure 2 also shows the microstructure changes occurring to the potato protein heat-induced gel (PoPI-G). Before digestion (Figure 2f), the microstructure consisted of a continuous and compact-dense network, characteristic of heat-induced protein gels. This gel, formed at neutral pH, is stabilized mainly by hydrophobic interactions [25,26], given the lack of sufficient internal disulfide bonds in the potato protein isolate. This would imply the extensive unfolding and exposure of non-polar moieties of the proteins, and make the proteins more prone to accessing the hydrolytic enzymes during digestion. Previous work has reported a denaturation temperature for patatin of 60.8 °C, while trypsin inhibitor gelation is induced after 70 °C [24,27]. In the present work, a denaturation peak for the commercial potato protein isolate was 68.7 ± 0.28, measured using differential scanning calorimetry (data not shown). Similar results were also found by Elahi and Mu [28]. During acidic gastric conditions (Figure 2g–i), the structure of the digesta transiting to the intestine still showed a compact network, with a brittle appearance and similar features between G1 and G3 emptying points (corresponding to a pH drop from 5.0–3.9). Only some large particles were visible at the final gastric point G5 (Figure 2i), and a homogeneous suspension of small aggregates was left at the final intestinal stage, I5 (Figure 2j).

The initial microstructure of PoPI-F was that of a typical foam (Figure 2k), containing large gas bubbles stabilized by a protein matrix, and with lamellae containing the remaining unadsorbed protein. The foam still retained its structure, at least in part, with the decrease of pH to 5.5 (Figure 2l), although the lamellae were thicker after the first stage of gastric digestion, due to dilution with eSSF, eSGF and the lower pH. However, the air bubbles were still quite visible and the protein particles seemed to be present in the interstitial phase as individual aggregates. This may indicate a difference in the structure of the proteins due to their surface denaturation. At G3 (pH 3.5) (Figure 2m), the air bubble structures collapsed, and an acid-induced protein-protein network around the lamellae was now visible, showing remnants of a gelled lamellar phase. At G5, it was no longer possible to collect the total calculated aliquot for each gastric emptying point (3.22 mL instead of 3.82 mL) as some of the foam structures persisted in the in vitro simulated gastric vessel. It is important to note that in vivo intervention studies have suggested the potential of aerated foods to increase gastric volumes, delay gastric emptying and reduce appetite [29]. The current in vitro study supports such findings, as it similarly shows the presence of significant phase inhomogeneities during digestion of a protein foam. Nevertheless, it is important to take into consideration parameters such as rheological properties and the stability of the foam, as well as the ability of the lamellae to coalesce during digestion, to determine the impact of these types of structure on gastric emptying and appetite reduction. A higher viscosity has been associated with a reduced gastric emptying rate of isocaloric and isovolumetric meals [30]. In order to characterize the rheology of the entire digesta, it has been recommended to apply oscillatory shear stress and a large deformation steady shear [31]. By the final gastric point, G5, the microstructure of what was emptied (Figure 2n) was quite similar to that of the PoPI-S with some residual particles. In addition, the final in vitro intestinal stage (Figure 2o) showed similarities to the unheated suspension.

The heated foam (PoPI-HF, Figure 2p) also showed a unique microstructure. In this case, the structure was very different than that of the unheated PoPI-F. There was a dense continuous protein network, but with trapped air bubbles. These air bubbles still showed round or ellipsoidal structures, suggesting that the PoPi created viscoelastic interfaces, able to withstand the heating process. The microstructure of the gastric emptied fractions (Figure 2q–s) in this case was somewhat similar to that of the heated PoPI-G. The structure was quite similar between G1 and G3 (Figure 2q,r), and large aggregates were still present at the final gastric stage, G5 (Figure 2s). The microstructure of the intestinal sample, I5 (Figure 2t), also shows more residual aggregates compared to the other treatments. The amount of gastric digesta emptied at each stage of the in vitro gastric digestion was also lower than that of the PoPI-F, as in this case, only approximately 3 mL could be emptied at each gastric emptying point, demonstrating a much higher resilience of this sample to digestion and emptying. It is possible to attribute this to the rheological properties and the resistance of this heat-set foam to gastric juices when compared to the unheated PoPI-F. These results are in full agreement with recent reports [32] suggesting that hunger suppression induced by aerated foods could be prolonged by enhancing foam stability in gastric conditions, as this increases gastric volumes and would decrease gastric emptying rates. For all the samples, the microstructural behavior of the different systems characterized by microscopy during the gastric and intestinal stages in the experiment was highly reproducible.

### 3.2. Sodium Dodecyl Sulphate-Polyacrylamide Gel Electrophoresis (SDS-PAGE)

With the aim of understanding the differences in the extent of protein breakdown between the four different isocaloric matrices prepared with potato protein, their non-digested state (U) as well as their digesta were analyzed at each in vitro gastrointestinal point (G1–G5, I1–I5) using SDS-PAGE (Figure 3). These samples represented the state of the protein transiting to the upper intestine at various digestion times. All samples shared a similar electrophoretic pattern at the initial stage (U), and their bands were assigned as reported in the literature [26,33]. Two main bands were present, a high molecular weight (Mw) protein (migrating at approximately 130 kDa) corresponding to lipoxygenase, and a second large band migrating at about 43 kDa corresponding to patatin. The protein isolate was mostly comprised of patatin; however low intensity bands were also noted for the trypsin inhibitor (TI) protein group (migrating at about 22, 15 and 9 kDa) (with two bands at 15 and 9 kDa). There were clear differences in the protein hydrolysis during gastrointestinal transit between the unheated and heated samples (Figure 3A–D). The PoPI-S and PoPI-F (Figure 3A,C) followed a similar gastric digestion pattern: the Mw of all band fractions remained unaltered from G1–G3, which indicated resistance to proteolysis by pepsin, yet due to the pH conditions they were non-optimal for the gastric enzyme’s activity [34]. However, at the two last emptying points, G4 and G5, a much higher extent of hydrolysis was noted, with only traces of patatin present, as well as some products of hydrolysis migrating at low Mw. At these two emptying points, conditions were ideal (highest pepsin concentration and pH ~2.0). At G5, a clear band appeared at 37 kDa, corresponding to pepsin, as the enzyme had reached high concentrations in the gastric digesta. It is important to note that when comparing G5 for PoPI-S and PoPI-F, there were some differences in the polypeptide distribution, with larger fragments left in the suspension, compared to the PoPI-F foam samples, and higher amounts of trypsin inhibitors still intact. Figure 3A,C also show the various stages of emptying after intestinal digestion (I1-I5) for PoPI-S and PoPI-F. There seemed to be more undigested protein in the suspension compared to the foam (Figure 3A,C), and the trypsin inhibitor bands were still visible at 15 kDa, in both suspension p0 and foam samples at the early stages of gastric emptying.

In the PoPI-G samples (Figure 3B), the protein seemed to be more resistant to digestion, with both lipoxygenase and patatin still visible at G4, and also at the final gastric stage (G5). Furthermore, in the PoPI-G samples, a new band (37 kDa) migrating under the native patatin fraction (43 kDa) appeared during gastric digestion G1–G4. This cannot be attributed solely to pepsin, as its concentration was not high enough to be detected at these early stages [35], and it may suggest that in the heated samples, patatin is subjected to a partial hydrolysis. As opposed to the two unheated PoPI structures (S, F) (Figure 3A,C), the heated gel still showed residual patatin at G5, as well as fragments migrating at low Mw. Previous studies have shown the resistance of native proteins to gastric digestion when compared to their denatured counterparts by using whey as a model protein [36,37]. This was attributed to peptide bonds in the native structure of whey proteins hindering the accessibility of the proteases [38]. This is not the case for potato protein isolate, where there was an apparent higher resistance to hydrolysis under gastric conditions compared to the unheated suspension. This may be attributed to the heat-induced denaturation of the protein, and/or to the physical structures formed in the gels. It has been previously suggested [39] that gel structures may provide a less effective medium for pepsin diffusion. The TI was present during the entire gastric digestion, and partly degraded under intestinal conditions, as shown by its migration at a lower Mw in Figure 3B.

The fractions with the largest peptide fragments transiting to the intestinal phase were noted for the heated foam. Figure 3D illustrates the protein distribution during in vitro digestion of the PoPI-HF. This treatment, as already suggested by the microstructural observations in Figure 2, showed the highest resistance to hydrolysis, during both the gastric and intestinal phase, demonstrating its persistence during gastrointestinal transit due to the structures created. In the initial stages of gastric digestion, patatin did not show changes, nor the appearance of the second band at lower molecular weight, which is shown in Figure 3B (PoPI-G). However, similarly to the gel, the patatin band was disrupted in the heat-set gel at the late stages of gastric digestion, with fragmentation of patatin visible only at G4 and G5, possibly due to a delay in digestion kinetics. Both PoPI-G and PoPI-HF showed a persistent patatin band at G5. The PoPI-HF also showed the largest population of partly degraded protein and TI during the latest stages of gastric digestion and in the intestinal stages. In particular, partly digested TI remained constant at all intestinal points (Figure 3D).

### 3.3. Determination of Trypsin Inhibitory Activity in PPI/Processing Effect on Trypsin Inhibitory Activity in Potato Protein Isolates

The residual trypsin inhibitory activity (TIA) present in the various PoPI processed samples was estimated (Table 1). All samples still showed residual trypsin inhibitory activity. The results support the findings described in Figure 3, in which TI bands were present in the undigested samples. The PoPI dispersion (72.8% ± 1.1) had the highest TIA, fully in agreement with the native character of the isolate. As expected, the unheated foam, PoPI-F (67.9% ± 2.9), also showed a high TIA, and no significant difference compared to the dispersion. Despite the drastic change in food structure, there was no surface denaturation of the TI. Heating treatment, on the contrary, showed a decrease in TIA; PoPI-G showed a slight reduction to 62% of TIA, and in the case of PoPI-HF, the reduction was significantly lowered to 25.6%. It has been previously shown that the structural change in protein natural conformation caused by heat-induced denaturation produced a non-reversible change, leading to a loss of inhibitory activity [40,41,42,43,44].

### 3.4. Peptide Size Distribution and Degree of Hydrolysis during Gastric Emptying and Intestinal Digestion

To follow the distribution of peptides formed during digestion depending on the matrix structure, the molecular weight distribution of the peptides present in the bioaccessible phase (centrifugal supernatants) was analyzed by size exclusion chromatography, for different gastric times (G1, G3, G5) as well as two intestinal stages (I1 and I5). Representative chromatograms are shown in Figure 4. I1 and I5 represent the simulated intestinal digestion of the first and last gastric emptying time (G1 and G5). The elution profile was separated into three different peptide populations, between 80 and 30 kDa, between 30 and 1 kDa, and <1 kDa, and the percentage over total elution area is shown in Figure 5. With the exception of the heat-set foam (Figure 5d), all the other PoPI treatments showed similar trends. The gastric digesta emptied at first had large Mw peptides (80–30 kDa), and this population then decreased; the small Mw peptides (<1 kDa), on the other hand, increased, reaching values larger than 80% of the total area during the intestinal digestion. This demonstrated that the supernatant fraction in the intestinal stage contained mostly bioaccessible peptides.

The Mw distribution of PoPI-S extracted in the soluble fraction at the early stages of gastric digestion consisted mostly of 80–30 kDa proteins, but this population no longer existed at the end of the gastric stage (G5) (Figure 5a). The proteolysis intensified with increasing pepsin concentration (constant addition, 48.42 μL min^−1^) and activity (due to the increasingly acidic environment), resulting in a complete shift of peptide distribution from G1 to G5, in which the 80–30 kDa group changed from 75.1% ± 1.6 to 1.9% ± 1.3, and the <1 kDa group changed from 4.0% ± 0.4 to 80.6% ± 7.4, making these peptides bioaccessible. It is important to note that even in the case of G1, the samples were fully digested once in the intestinal phase, as shown by I1.

When comparing the peptide population in the soluble extract between PoPI-S and PoPI-G, the heat-treated counterpart, it was clearly shown that the high Mw peptides in PoPI-G were significantly lower (*p* < 0.05) in G1, by 25%, and in G3, by 45%, while the low Mw peptide group was significantly higher (*p* < 0.05) in G1, by 29%, and in G3, by 44%. This would indicate that the breakdown of proteins was faster in the heated gel structure, compared to the unheated suspension. However, it is important to note that these were the peptides present in the soluble fraction. Indeed, the SDS-PAGE data (Figure 3) suggests a higher resistance to digestion of the proteins in the heated PoPi-G and PoPi-HF.

As already shown in Figure 3, PoPI-F followed a proteolysis trend similar to that shown in all gastrointestinal stages of PoPI-S. However, there was a higher percentage of the 80–30 kDa group in PoPI-F than PoPI-S at the G1 and G3 points. No significant differences were found in the other stages. Eventually, foam decay by liquid drainage and further collapse of the air bubbles (coalescence and Ostwald ripening) occurred, which led to a similar digestive behavior between the suspension and the foam. Both structures are examples that confirm the synergistic interaction of pepsin and trypsin to hydrolyze protein. A study by Rivera del Rio and colleagues [45] previously demonstrated the increased efficiency in hydrolysis when both enzymes were present under ideal conditions, signaling that gastric hydrolysis facilitated the trypsin hydrolysis of a model protein (Bovine Serum Albumin) in the small intestine. The increase in small peptide distribution (<1 kDa) from I1 (59.9% ± 1.8) to I5 (81.9% ± 2.9) in PoPI-F as well as a similar increase in the same group for PoPI-S confirm the previous findings.

In the case of PoPI-HF, very little high Mw peptides were present in the bioaccessible fraction, even at G1. This may indicate a very low rate of dissolution of the protein at the initial stages of digestion. At G1, the peptide distribution pattern of this food structure already consisted mostly of low Mw peptides (46.5% ± 6.0). Despite the continuous addition of pepsin and gastric juices, the low Mw peptide group decreased continuously in G3 and G5 (35.5% ± 5.3 and 26.2% ± 0.8, respectively). The slow protein breakdown from G1 to G5 was attributed to the high stability and resistance of the heat-set foam, as the structure may have hindered the diffusion of gastric juices in the vessel. This resulted in a delayed and heterogeneous digestion process, which resulted in limited gastric emptying. This experimental set-up was considered relevant to in vivo conditions as HCl is released from parietal cells, which are present in the glands within the fundus and body of the stomach (from the top) [46], while mixing intensifies in the antrum [39]. Despite the structure resistance to gastric breakdown and slow gastric emptying, once the suspensions were transferred to the intestinal stage, the digestion was immediate. Both fractions emptied at G1 and G5 showed a high release of low Mw peptides after the intestinal stage (I1, I5), again demonstrating the prompt digestion of the potato protein isolates.

To further determine the impact of pepsin and trypsin during the gastrointestinal points of each PoPI structure, as well as the effect of enzymatic interaction, the degree of hydrolysis (DH%) was studied. The extent of protein hydrolysis in all the gastrointestinal digestion points was evaluated by measuring the free amino groups released and expressed as g of L-Leucine equivalents per total amount of protein in the initial sample (40 g/L), as shown in Figure 6. All samples showed a very low degree of hydrolysis (DH) (less than 10%) regardless of gastric emptying times. However, the samples showed a higher degree of hydrolysis after intestinal digestion (≥51.2%). DH was slightly lower in I1 compared to the later points, confirming that an increase in pepsin concentration and gastric incubation time ultimately favored trypsin hydrolysis. Differences in DH kinetics were marked by the matrix structure of the heated foams, which showed lower values compared to the PoPI-G, also heated.

Among all the samples, the suspension had the highest release of free NH_2_ groups (5.8% ± 0.3) under early gastric conditions (G1), while at this same time point, the other treatments presented values ~1%. These values slightly increased at the later gastric stages. At the last gastric emptying point, the suspension, gel and foam had similar DH values (~8%), while PoPI-HF only reached 3.2% ± 0.8. Thus, the differences in gastric digestion between the four PoPI structures could be due to the influence of matrix structure complexity on pepsin diffusivity in the sample [47].

As expected, all samples showed a drastic increase in DH from gastric to their respective intestinal points due to efficient activity by trypsin and chymotrypsin. As shown in Table 1, there were trypsin inhibitor activities for all samples, with the lowest residual activity being that of the heated foams. However, it is important to note that the levels of TIA were estimated to range from 125 U (in the sample) in the PoPI-S to 44 U in the PoPI-HF, and were much lower than the amount of trypsin present in the simulated intestinal fluid (764 U). Despite having a lower TIA, PoPI-HF did not show a higher DH than PoPI-G, confirming the hypothesis that the structuring in the gut influenced protein digestibility.

Remarkable differences between some of the PoPI structures appeared not only at the same point (e.g., suspension vs. gel), but also between the same structure at different intestinal points (e.g., I1 vs. I5). For instance, the suspension (PoPI-S) at I1 (66.5% ± 4.3) had a significantly higher DH than the unheated foam (51.2% ± 0.6) (Figure 4A). Nevertheless, both structures showed similar values from I2 to I5, reaching a final DH of about 75%. The released amount of free NH_2_ groups by these two unheated treatments, regardless of the structure, showed an increase at the initial stages of gastric emptying, confirming the essential role of pepsin activity in increasing digestibility in the intestinal stages. It is also clear that the presence of native trypsin inhibitor did not have an impact on the hydrolysis of these samples.

Figure 6B shows the DH for the heated samples, the gel and the heated foam. PoPI-G reached the highest DH values during the simulated intestinal stage. This treatment showed fast hydrolysis kinetics, as even for the first gastric emptying stage, the values already reached 83.4% ± 1.3, and continued to grow to 96.6% ± 4.4. Despite reporting the lowest TIA, PoPI-HF had a similar DH to those of the unheated suspension. Although the gastric emptying was inhibited by the heated foam structure, once the heated protein passed to the intestinal stage, there was an increasingly higher level of DH for each gastric step, once again confirming the important role of pepsin in imparting the digestibility of the samples.

### 3.5. Free Amino Acid Analysis by Triple Quadrupole Mass Spectrometry

Individual free amino acids (FAA) released were studied using LC/TQMS at the first (I1) and last intestinal point (I5) to evaluate differences in the composition of the bioaccessible fractions for the four different potato matrices. The FAA values at I5 are reported in Table 2.

PoPI-S showed the highest release of FAA (Table 2), while the other three samples had similar values. This is a consequence of structural differences in the matrix, as PoPI-S rapidly releases FAA, due to its high solubility. There was no difference in the ratio of individual FAA released and values were similar among all the structures, indicating a similar digestion of the protein fractions within the treatments. Methionine was the essential amino acid with the lowest release at the last intestinal point for all PoPI matrices, followed by arginine. The limiting amino acid in potato protein sources is often considered to be methionine [48] while leucine is the limiting factor for colored potatoes [6,49]. The results of the current study would point to methionine and valine as the limiting amino acids in the commercial potato protein isolate.

Gorissen and collaborators [1] reported a high level of BCAA in the composition of a commercially available potato protein source. The present study confirmed these results by demonstrating that PoPI released high amounts of branched chain amino acids (BCAA: valine, leucine and isoleucine) under intestinal conditions, amino acids responsible for muscle protein synthesis, fatigue recovery and decrease in exercise-induced muscle damage [50,51]. In our study, the values of released FAA were also high for lysine, phenylalanine, threonine and tryptophan.

Different studies have shown the high quality of potato protein, reporting high protein digestible corrected amino acid scores and digestible indispensable amino acid scores (PDCAAS = 1.0, DIAAS = 0.94, 1.00) [52,53,54], as it contains all essential amino acids in sufficient quantities [54]. However, the recently claimed DIAAS for potato protein reported in the literature was based on the initial amino acid composition of the product and not on the ileal bioaccessible fraction. This is of critical importance, as protein digestibility is overlooked or assumed to be 100%, which results in a predictive DIAAS value. This work calculates the DIAAS value in vitro after digestion. Based on the free amino acids quantified from the potato protein isolate (suspension) at the intestinal endpoints, I5, using the in vitro semi-dynamic INFOGEST system, an in vitro DIAAS was calculated in our study and reported, as shown in Table 3.

## 4. Conclusions

In this study, semi-dynamic in vitro digestion proved to be a useful system that not only provided information about the effect of ideal gastrointestinal conditions on protein bioaccesibility, as would the static in vitro system, but also allowed a mechanistic understanding of the effect of structural differences on the entire gastrointestinal digestive process. The obtained results using this system suggest that food protein matrices with isocaloric content could present differences in gastric emptying dynamics when considerable structural differences occur. Thus, a greater structural complexity could lead to more delayed gastric emptying, impacting the dynamics of gastrointestinal digestion. These results need to be further tested under in vivo conditions. The use of MRI could provide better insights and allow for validation of the in vitro semi-dynamic model.

Electrophoretic patterns showed visual differences in protein breakdown, especially in patatin bands during gastric points and in trypsin inhibitors under intestinal conditions. Trypsin inhibitory activity was greatly reduced by heat treatments; nevertheless, there was no substantial increase in the amount of free amino groups released as measured by OPA assay, when compared to non-heat-treated potato protein structures. This is mainly due to the low presence of TI in commercial potato protein ingredients. Lastly, an in vitro DIAAS for the potato protein isolate was calculated and the result indicates that the product is a high quality protein (DIAAS = 87). More in vivo research is still needed in order to determine the effect of consuming low amounts of trypsin inhibitors in the long term. Due to its denatured state, the gel structure not only had the highest degree of hydrolysis but the highest release of free amino groups in the early stages of intestinal digestion. Due to the easy gastric emptying, the suspension had a high release of free amino acids, but not the highest degree of hydrolysis. Contrarily, more complex structures in the heat-set foam had a higher degree of hydrolysis in the last intestinal point but a lower release of free amino acids. Thus, this study highlights the importance of applying both degree of hydrolysis by the OPA method and free amino acids by LC/TQMS in order to deepen understanding of the effect of complex structures on protein digestive mechanisms.

It is possible that this type of protein ingredient could find applications as a “slow protein” and have an impact on the modulation of postprandial metabolism. The incorporation of this ingredient in food matrices could favor the development of certain dietetic foods with low caloric contents. This work not only explored the effect of the structuring of potato protein isolates on the gut and its impact on hydrolysis kinetics, but also highlighted the importance of adequate processing conditions of refined ingredients obtained from by-products of alternative protein sources, as this highly functional ingredient has the potential to be successfully used in novel food matrices.

## Figures and Tables

**Figure 1 nutrients-14-02505-f001:**
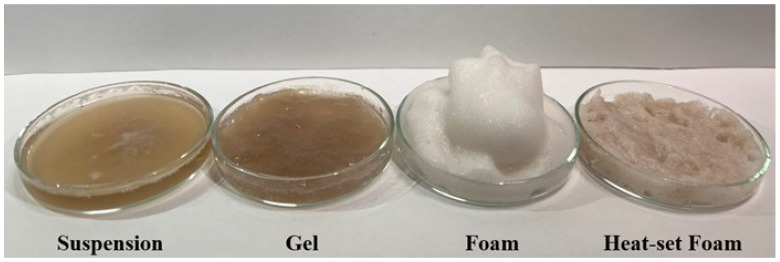
Visual appearance of potato protein isolate structures (4% protein, *w*/*w*).

**Figure 2 nutrients-14-02505-f002:**
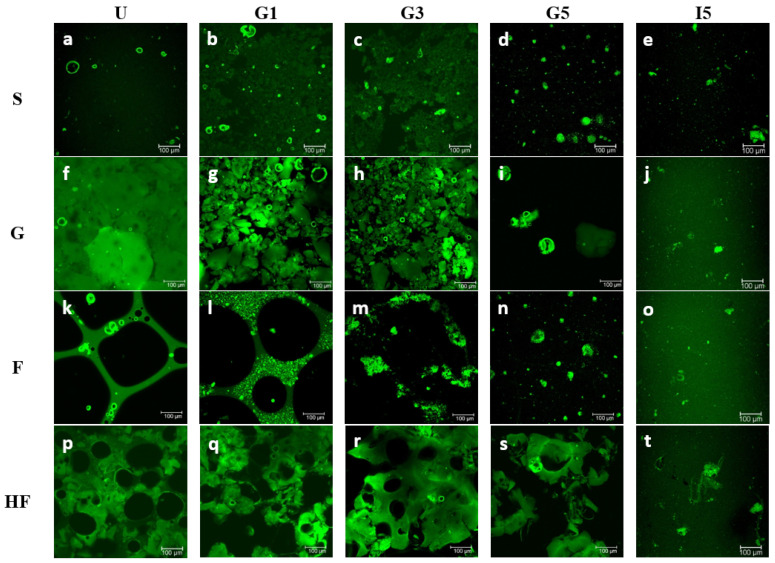
Representative confocal scanning laser microscopy (CSLM) images of four different potato protein isolate structures (4% protein, *w*/*w*). Structures: Suspension (S) (figures **a**–**e**), Gel (G) (figures **f**–**j**), Foam (F) (figures **k**–**o**), Heated Foam (HF) (figures **p**–**t**). Different gastric endpoints shown: GE1 (8.26 min), GE3 (24.78 min), and GE5 (41.30 min). Scale bar: 100 μm.

**Figure 3 nutrients-14-02505-f003:**
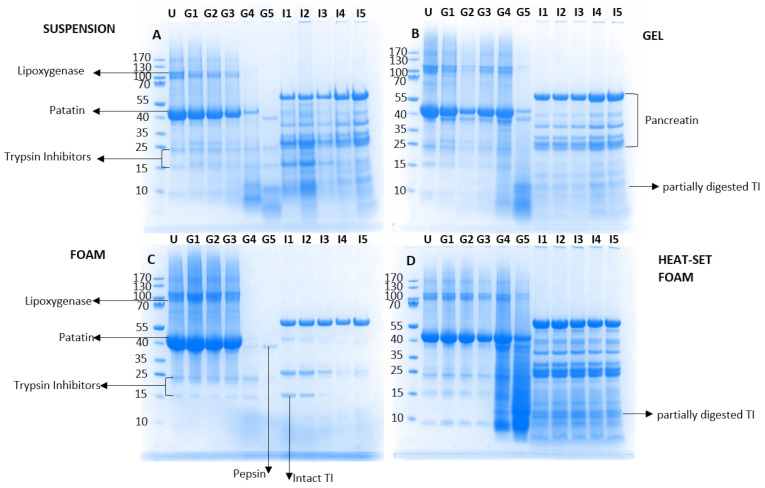
SDS-PAGE profiles of four different potato protein isolate matrices (4% protein, *w*/*w*) under reducing conditions: suspension (**A**), gel (**B**), foam (**C**), heat-stable foam (**D**). Undigested (U), and at the following gastric emptying points: G1 (8.26 min), G2 (16.52), G3 (24.78 min), G4 (33.04 min), G5 (41.30 min). On the right-hand side of each gel, the samples loaded are the corresponding intestinal points of each gastric point (I1–I5). Molecular weight standards are indicated on the left side, and he tmain protein fractions identified are indicated with arrows.

**Figure 4 nutrients-14-02505-f004:**
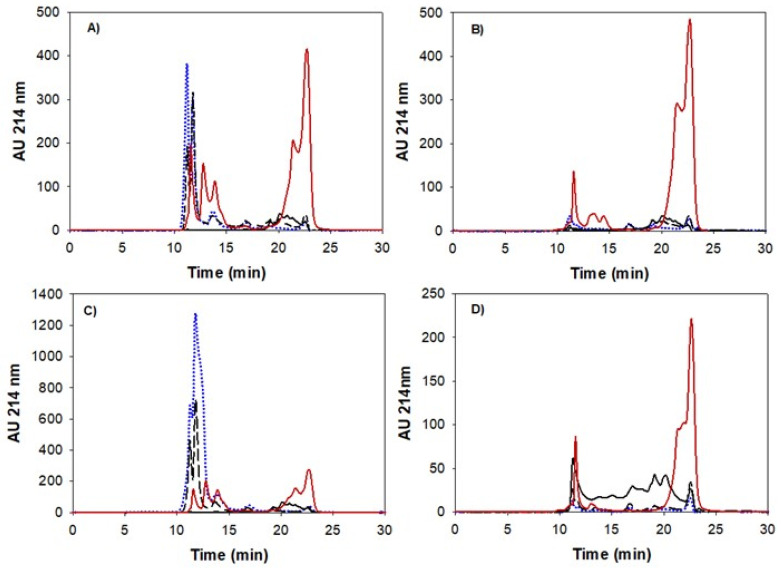
Representative elution patterns for supernatants of the digestates at G1, G3, G5 and I1 analyzed by size exclusion chromatography (see methods) for potato protein isolates: suspension (**A**), gel (**B**), foam (**C**), heated foam (**D**). Standard peaks elution: 100 kDa–10.6 min, 30 kDa–13 min, 1 kDa–19 min. G1 (dotted blue line), G3 (broken line), G5 (continuous black line) and intestinal, corresponding to G1 (red line).

**Figure 5 nutrients-14-02505-f005:**
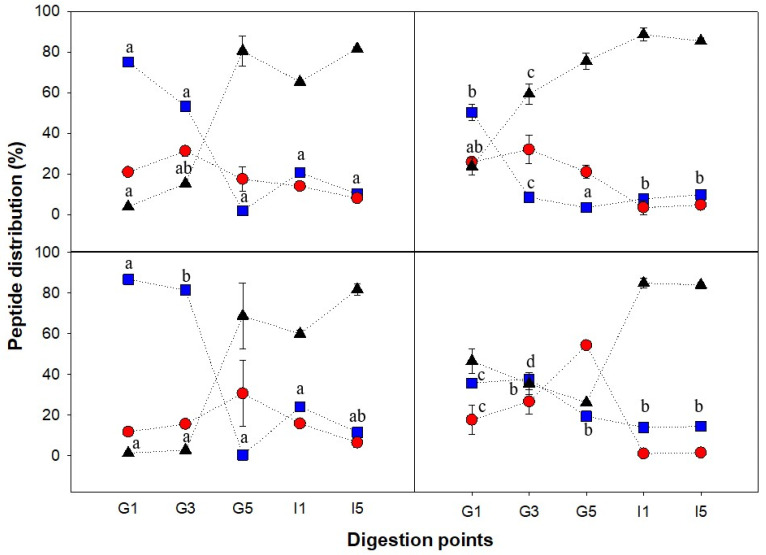
Molecular weight distribution of the peptides present in the soluble fraction of the four potato protein isolate (PoPI) samples. Suspension (a), gel (b), foam (c), heat-set foam (d). Three different Mw ranges are presented: 80–30 kDa (squares), 30–1 kDa (circles), <1 kDa (triangles). Bars represent standard deviation. Letters indicate differences within the same Mw population. Lines are drawn to guide the eye. Gastric stages: G1 (8.26 min), G3 (24.78 min), G5 (41.30 min), and corresponding intestinal points (I1 and I5).

**Figure 6 nutrients-14-02505-f006:**
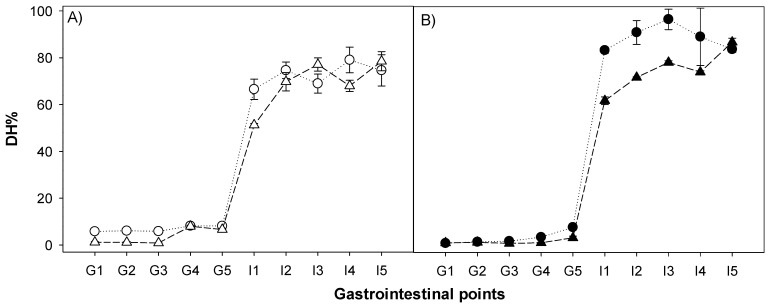
Release of free amino groups in the different potato protein isolate treatments. (**A**) Unheated treatments, suspension (PoPI-S, empty circle), foam (PoPI-F, empty triangle); (**B**) heated treatments, gel (PoPI-G, filled circle), heat-set foam (PoPI-HF, filled triangle). Gastric emptying points: G1, G2, G3, G4, G5 and intestinal points corresponding each gastric point (I1, I2, I3, I4 I5). Dotted lines are used to guide the eye.

**Table 1 nutrients-14-02505-t001:** Trypsin Inhibitory Activity (TIA, Units/mg solids) and percentage of inhibition (TIA%) (see methods) for various treatments. One unit of TIA produced 50% inhibition of 2 Trypsin units. Values are reported as the average of three replicates with standard deviation. Different letters indicate statistically significant differences between treatments.

Trypsin Inhibitory Activity	PoPI-S	PoPI-G	PoPI-F	PoPI-HF
TI (U/mg PoPI)	3.1 ± 0.0 ^a^	2.7 ± 0.0 ^b^	2.9 ± 0.1 ^a^	1.1 ± 0.0 ^c^
TIA (%)	72.8 ± 1.1 ^a^	61.9 ± 1.2 ^b^	67.9 ± 2.9 ^a^	25.6 ± 0.3 ^c^

**Table 2 nutrients-14-02505-t002:** Free amino acid values obtained by LC/TQ-MS measured after in vitro intestinal digestion (I5). Unheated suspension (PoPI-S), gel (PoPI-G), foam (PoPI-F), heat-set foam (PoPI-HF). The values are expressed as mg of free amino acid per g soluble potato protein in the intestinal point (I5). Values are reported as average of two replicates with standard deviation. The amount of total free amino acids (TFAA) is also shown: total free amino acids. Values highlighted in bold correspond to limiting amino acids (lysine and histidine). Different letters indicate statistically significant differences between treatments for a given amino acid.

mg Free Amino Acids/g Soluble Protein	PoPI-S	PoPI-F	PoPI-G	PoPI-HF
Essential amino acids				
Valine	33.5 ± 0.1 ^a^	33.3 ± 7.3 ^a^	24.2 ± 1.1 ^a^	23.4 ± 0.5 ^a^
Leucine	78.5 ± 2.4 ^a^	46.8 ± 15.8 ^a^	55.5 ± 0.2 ^a^	58.5 ± 2.4 ^a^
Isoleucine	119.1 ± 1.3 ^a^	81.2 ± 28.0 ^a^	97.8 ± 3.6 ^a^	96.0 ± 2.5 ^a^
Methionine	9.9 ± 0.1 ^a^	6.0 ± 2.0 ^a^	6.9 ± 0.05 ^a^	7.4 ± 0.16 ^a^
Phenylalanine	43.3 ± 2.1 ^b^	25.5 ± 8.0 ^a^	38.9 ± 1.2 ^ab^	28.8 ± 2.2 ^ab^
Threonine	30.3 ± 0.5 ^a^	20.6 ± 7.5 ^a^	23.5 ± 1.13 ^a^	23.3 ± 0.3 ^a^
Lysine	92.5 ± 2.1 ^b^	56.1 ± 19.7 ^a^	66.8 ± 1.6 ^a^	67.6 ± 1.7 ^a^
Histidine	45.4 ± 1.77 ^b^	32.2 ± 11.7 ^a^	31.7 ± 2.2 ^a^	36.8 ± 3.2 ^a^
Tryptophan	35.7 ± 0.9 ^a^	23.5 ± 7.3 ^a^	31.6 ± 0.8 ^a^	27.2 ± 1.2 ^a^
Non-essential amino acids				
Tyrosine	30.4 ± 0.2 ^a^	19.7 ± 6.3 ^a^	26.3 ± 0.4 ^a^	26.0 ± 0.9 ^a^
Asparagine	17.2 ± 0.4 ^a^	41.8 ± 6.2 ^b^	21.4 ± 0.3 ^a^	27.6 ± 2.3 ^ab^
Cysteine	10.0 ± 0.1 ^ab^	13.5 ± 0.4 ^c^	10.4 ± 0.2 ^b^	9.1 ± 0.07 ^a^
Proline	2.48 ± 0.0 ^a^	1.74 ± 0.7 ^a^	1.6 ± 0.03 ^a^	1.4 ± 0.06 ^a^
Glycine	17.1 ± 4.1 ^a^	25.6 ± 3.2 ^a^	15.8 ± 0.6 ^a^	16.7 ± 0.6 ^a^
Glutamic acid	69.7 ± 0.2 ^a^	38.7 ± 16.0 ^a^	47.6 ± 0.2 ^a^	44.4 ± 2.0 ^a^
Aspartic acid	172.2 ± 12.6 ^a^	102.8 ± 42.2 ^a^	119.5 ± 1.3 ^a^	116.1 ± 7.2 ^a^
Alanine	58.2 ± 0.8 ^a^	35.0 ± 11.4 ^a^	43.9 ± 0.4 ^a^	40.8 ± 0.2 ^a^
Serine	15.4 ± 1.3 ^a^	10.4 ± 4.9 ^a^	14.18 ± 1.1 ^a^	11.9 ± 1.0 ^a^
Glutamine	64.1 ± 3.7 ^a^	46.1 ± 14.6 ^a^	45.7 ± 2.3 ^a^	42.6 ± 4.4 ^a^
Arginine	0.1 ± 0.0 ^a^	0.0 ± 0.0 ^a^	0.0 ± 0.0 ^a^	0.0 ± 0.0 ^a^
TFAA	1000 ± 35 ^b^	695 ± 224 ^a^	762 ± 20 ^a^	743 ± 34 ^a^

**Table 3 nutrients-14-02505-t003:** Digestible indispensable amino acid score (DIAAS) calculations for potato protein based on initial amino acid composition of the potato protein sample [53] and DIAAS based on the in vitro digestibility by measuring presence of essential individual essential amino acids per g of protein (mg/g) in the bioaccessible fraction available at the ileum of the last intestinal point (I5). Free amino acids were measured by LC-TQ-MS. * Values of the reference pattern correspond to the adult maintenance amino acid pattern (mg/g) as recommended in the dietary protein quality evaluation in human nutrition report (FAO, 2013).

Essential Amino Acids	Amino Acid Content in the Reference Pattern Values (mg/g) *	Potato Protein Isolate Free Amino Acids Released Ileum (mg/g). Experimental Data	Theoretically Predicted DIAAS Based on Initial Amino Acid Composition of the Source (Before Digestion) [53]	Calculated DIAAS Based on INFOGEST In Vitro Experimental Bioaccesible Values (In Vitro DIAAS Adults)
Valine	39	34	1.38	0.87
Leucine	59	78.5	1.43	1.33
Isoleucine	30	119.12	1.56	3.97
Met + Cys	22	20	1.15	0.91
Phe + Tyr	38	73.7	2.10	1.93
Lysine	45	92.5	1.22	2.05
Threonine	25	30.4	1.65	1.22
Tryptophan	6.6	35.7	1.28	5.41

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
