# Peer review of "Impact of the Structural Modifications of Potato Protein in the Digestibility Process under Semi-Dynamic Simulated Human Gastrointestinal In Vitro System"

_nutrients, 2022, doi:10.3390/nu14122505_

Round 1
Reviewer 1 Report
The title should be concise and specific, right now the title is too extensive and rather confusing.
Suggestion: "Impact of the structural modifications of potato protein in the digestibility process under semi-dynamic simulated human gastrointestinal in-vitro system"
Please make sure to follow the manuscript guidelines. The Abstract must not exceed 200 words.
L37. Is it "plant protein matrices" or food matrices containing plant proteins?
L44-46. Please add references to substantiate these claims.
L64-67. CMC complexation and salt precipitation are recovery techniques that could be mentioned as well.
L84. Please elaborate a bit more about the in-vitro system used.
L90. It would be preferable to expose the overall formulation of each structure developed.
L253. I found the discussion a bit confusing and the manuscript would certainly beneficiate from a more integrated discussion.
R&D
I suggest that the authors add pictures of the different PoPi systems developed (suspension, gel, unstable and stable foam).
L301-302. Please rephrase the sentences. This part is a bit confusing.
L313. Substitute for "inhomogeneities". Is the ordering correct?
L314. What is the author's suggestion in regard to this? Can you elaborate on the approach that could be used to target such?
What was the level of reproducibility regarding the microstructural behaviour of the different systems during gastric and intestinal stages?
L335. On the other side, at which level will excessive aeration be detrimental for consumers?
Conclusions:
Do you think is possible to elaborate on the suitability of the different systems (according to their digestive behaviour) to be included in specific food products? Can you expand a bit on the subject?
Author Response
- The title should be concise and specific, right now the title is too extensive and rather confusing. Suggestion: "Impact of the structural modifications of potato protein in the digestibility process under semi-dynamic simulated human gastrointestinal in-vitro system"
Answer: We thank the reviewer for the suggestions and we agree on the recommendation, thus, the title was changed to the proposed one by the reviewer in order to make more clear and specific.
- Please make sure to follow the manuscript guidelines. The Abstract must not exceed 200 words.
Answer: The abstract has been adjusted to l92 words.
- Is it "plant protein matrices" or food matrices containing plant proteins?
Answer: Thank you for the suggestion, “plant protein matrices” (previously L37) has now been replaced for “food matrices containing plant proteins” (now L32).
- L44-46. Please add references to substantiate these claims.
Answer: References supporting such claims have now been added
- L64-67. CMC complexation and salt precipitation are recovery techniques that could be mentioned as well.
Answer: The techniques suggested are now mentioned (L60).
- Please elaborate a bit more about the in-vitro system used.
Answer: A phrase explaining why the semi-dynamic in vitro system used was chosen has now been added. (L79).
- It would be preferable to expose the overall formulation of each structure developed.
Answer: The structures were all composed of 4% protein. There was nothing else added. No changes were made to the document.
- I found the discussion a bit confusing and the manuscript would certainly beneficiate from a more integrated discussion.
Answer: We thank the reviewer for the insight, we agreed and now modifications to the discussion of the microstructure section have been made. This in order to make it more fluid, concise and easy to understand.
R&D
- I suggest that the authors add pictures of the different PoPi systems developed (suspension, gel, unstable and stable foam).
Answer: A picture showing the PoPI matrices has been added to the manuscript.
- L301-302. Please rephrase the sentences. This part is a bit confusing.
Answer: The sentence has been rephrased and made clearer.
- Substitute for "inhomogeneities". Is the ordering correct?
Answer: misspelling has now been corrected. (now L309).
- What is the author's suggestion in regard to this? Can you elaborate on the approach that could be used to target such?
Answer: A new paragraph has been added elaborating on the importance of rheology and the recommended approach. (L313-316).
- What was the level of reproducibility regarding the microstructural behaviour of the different systems during gastric and intestinal stages?
We thank the reviewer for this question, it is indeed something worth mentioning. This has now been added at the end of the discussion of the microstructure section: “For all the samples, the microstructural behavior of the different systems characterized by microscopy, during gastric and intestinal stages in the experiment, was highly reproducible” (L337).
L335. On the other side, at which level will excessive aeration be detrimental for consumers?
Answer: This is an interesting question, but we do not think we need to add an additional comment to this document, as this paper is about structure differences. Foams are structures of interest for foods, and most probably not a vehicle for excessive air ingestion by consumers. Regardless it was not within the scope of this work to test such hypothesis.
Conclusions:
Do you think is possible to elaborate on the suitability of the different systems (according to their digestive behaviour) to be included in specific food products? Can you expand a bit on the subject?
Answer: The phrase “The incorporation of this ingredient in food matrices could be favor the development of certain dietetic foods with low caloric content” in the conclusion section (L646) addressing this question has now been added.
Reviewer 2 Report
This is a nice and elegant work demonstraaating the importance of the food preparation on the protein digestion and , therefore, in the aminoacid uptake.
In my opinion, the figure 5 is not representative and should be replaced by the chromatograms presented in the suplementary files.
On the other hand, the PCA analysis is incomplete and have some drawbacks that must be reviewed. Considering the set of data, I suggest to remove the PCA analysis since it does not agregate new information to the work. However, if the authors chose to maintain the PCA results, please provide the complete table of correlation (Pearson correlation) and please provide a complete analysis of the PCA1 and PCA2.
Author Response
This is a nice and elegant work demonstraaating the importance of the food preparation on the protein digestion and , therefore, in the aminoacid uptake.
- In my opinion, the figure 5 is not representative and should be replaced by the chromatograms presented in the suplementary files.
Answer: We thank the reviewer for the valuable insights. Figure 5 has been moved to the supplementary material. The figure has been replaced, and the chromatograms previously presented in the supplementary material are now present in the manuscript.
- On the other hand, the PCA analysis is incomplete and have some drawbacks that must be reviewed. Considering the set of data, I suggest to remove the PCA analysis since it does not agregate new information to the work. However, if the authors chose to maintain the PCA results, please provide the complete table of correlation (Pearson correlation) and please provide a complete analysis of the PCA1 and PCA2.
Answer: The PCA analysis has been moved to the supplementary material.